# Hybrid Modeling of Serious Vaccine Adverse Events Using Narrative Embeddings and Structured Data

Jonathan Feldman
*Glen Ridge High School*
Glen Ridge, New Jersey, USA
jonathan.feldman.2007@gmail.com

*Abstract*—We present a pipeline that combines structured metadata with sentence-level embeddings from narrative symptom text in VAERS to predict serious vaccine outcomes. Using pretrained language models such as SBERT, BioBERT, and ClinicalBERT, we achieve improved classification accuracy compared to models that rely solely on structured data. SBERT performs best overall, while ClinicalBERT also performs well, demonstrating the value of both general-purpose and domain-specific models. Clustering the symptom embeddings reveals clinically relevant patterns, such as shingles flare-ups and neurological issues, that may not appear in structured fields. Our findings suggest that incorporating narrative text enhances both predictive accuracy and interpretability.

*Index Terms*—Vaccine safety, NLP, Sentence-BERT, BioBERT, ClinicalBERT, adverse event detection, symptom clustering, pharmacovigilance

## I. Introduction

VAERS [1] is a key system for monitoring vaccine safety in the U.S. after approval. It contains over 2.6 million reports from patients, clinicians, and public health officials describing post-vaccination reactions. These reports include both structured metadata (e.g., age, sex, vaccine type) and unstructured narrative descriptions of symptoms and clinical outcomes.

While structured fields are commonly used for modeling and signal detection, the narrative text remains underused. The narratives often include important details such as symptom timing, severity, and progression – information that is often missing from coded fields.

In this work, we present a scalable framework that combines structured metadata with sentence-level embeddings from narrative text to improve vaccine safety analysis. We analyze VAERS reports spanning over three decades using transformer-based models (SBERT, BioBERT, ClinicalBERT) for two main tasks: predicting serious outcomes and clustering symptom patterns.

We show that combining structured data with narrative embeddings improves prediction performance across multiple metrics. We also use clustering to identify symptom patterns, such as shingles flare-ups and neurological issues, that are linked to serious outcomes but may not be visible in structured fields. In addition, we explore how these clusters vary across vaccine brands.

Although we use existing models, we apply them in a new way by combining them with structured data at scale. To our knowledge, this is the first study to evaluate this type of hybrid modeling on VAERS for both classification and exploratory analysis. Our results suggest that integrating narrative text can improve both prediction and interpretability in vaccine safety monitoring.

## II. Related Work

Post-market surveillance of vaccine safety using VAERS has traditionally focused on structured variables such as age, sex, vaccine type, and coded symptom fields [2], [3]. Analytical methods typically include disproportionality analysis, signal detection algorithms, and regression-based modeling. These approaches rely heavily on structured data and largely overlook the narrative symptom text.

Early efforts to analyze VAERS narratives used keyword matching, co-occurrence patterns, or unsupervised techniques like TF-IDF clustering and Latent Dirichlet Allocation (LDA) [4]–[6]. However, such methods often produced clinically imprecise or ambiguous groupings [7].

Recent advances in language modeling have enabled more nuanced representations of clinical narratives. Transformer-based encoders such as Sentence-BERT (SBERT) [8], BioBERT [9], and ClinicalBERT [10] have been applied to biomedical applications ranging from adverse drug event extraction to patient stratification [11]. For instance, Du et al. developed a BERT-based system to extract named entities related to Guillain-Barré syndrome from VAERS narratives, demonstrating the utility of deep learning in adverse event recognition [12].

More recently, von Csefalvay introduced DAEDRA, a domain-adapted language model for predicting vaccine-related outcomes using narrative text alone [13]. While promising, this work focused exclusively on unstructured inputs and did not incorporate structured metadata or explore hybrid modeling strategies. In contrast, our method integrates both modalities and provides detailed ablation and interpretability analyses. Tiwari et al. proposed a classification pipeline using NLP and character/word-level embeddings to predict hospitalization from COVID-19 vaccine symptoms [14], but their models did not include structured features or clustering analysis.

Several other studies have applied transformer embeddings to VAERS for unsupervised or descriptive tasks. Wang et al. used text embeddings and ontologies to analyze vaccine response patterns [15]; Cheon et al. clustered COVID-19 vaccine narratives using unsupervised learning [16]; and Li et al. examined temporal and spatial trends in structured and narrative fields [17]. These works provide important insights into data structure and signal emergence, but they do not evaluate predictive models for case severity.

In contrast, our study explores the integration of narrative embeddings and structured metadata for supervised classification of serious vaccine outcomes. While we use existing pretrained encoders, we focus on evaluating the added value of narrative context in improving predictive performance. We further examine symptom-level structure through clustering and validate model behavior using token-level perturbation analysis to support transparency in surveillance applications

## III. DATA AND METHODS

### A. Data Source and Preprocessing

We use the publicly available VAERS dataset [1], which includes all U.S. reports from 1990 through early 2025. Each record contains structured fields (e.g., age, sex, vaccination date, vaccine brand) and a narrative describing symptoms and timelines of adverse events. The dataset spans over 2.6 million reports across a wide range of vaccine types, including COVID-19, influenza, shingles, HPV, and hepatitis.

We merged three core VAERS tables (`VAERSDATA`, `VAERSSYMPTOMS`, and `VAERSVAX`), using the unique `VAERS_ID`. Reports with missing or empty narratives were excluded from text-based analysis but retained for structured-only models.

To reduce noise, we removed (1) narratives with fewer than five tokens and (2) near-duplicate entries with cosine similarity above 0.98, using SBERT embeddings. This filtering step removed 170,320 short and 172,106 duplicate reports, resulting in a final dataset of 1,402,333 narrative reports for embedding and downstream analysis.

All text was lowercased and stripped of punctuation, numbers, and extra whitespace before embedding. Embeddings were later computed using SBERT [8], BioBERT [9], and ClinicalBERT [10].

### B. Symptom Embedding and Clustering

We used three sentence embedding models: SBERT (`all-MiniLM-L6-v2`), BioBERT (`S-BioBERT-snli-multinli-stsb`), and ClinicalBERT (`Bio_ClinicalBERT`). SBERT outputs 384-dimensional vectors; BioBERT and ClinicalBERT output 768-dimensional vectors. SBERT is trained on semantic similarity tasks, while the others are pretrained on biomedical and clinical texts.

Embeddings were generated in mini-batches of 512 using GPU acceleration. KMeans clustering ($k = 10$) was applied to each model's embeddings. We selected $k$ based on silhouette

scores and manual review of cluster coherence. Each report was assigned to one cluster.

To interpret clusters, we examined representative narratives and used TF-IDF centroids to extract key terms [18]. UMAP projections were generated to visualize cluster separability.

We selected $k = 10$ based on qualitative coherence of the clusters and interpretability of resulting symptom groupings. Figure 1 compares KMeans clustering outputs from all three embedding models using UMAP projections, highlighting differences in cluster geometry and interpretability.

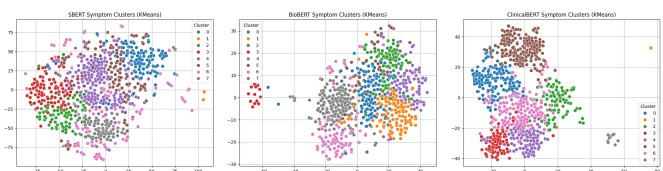

Fig. 1: KMeans clustering of VAERS symptom narratives using SBERT (left), BioBERT (center), and ClinicalBERT (right) embeddings. Each point is a report, colored by assigned cluster. ClinicalBERT produced tighter geometric groupings, but SBERT clusters were more clinically distinct based on manual review. SBERT was selected for downstream analysis due to interpretability and inference speed.

### C. Integration with Structured Data

We trained gradient-boosted decision trees using XG-Boost [19]. Structured features were concatenated with sentence-level embeddings from SBERT, BioBERT, or ClinicalBERT to form hybrid input vectors. We used 5-fold stratified cross-validation and evaluated performance using AUC, precision, recall, F1, and average precision. We compared four configurations: (1) structured-only, (2) structured + SBERT, (3) structured + BioBERT, and (4) structured + ClinicalBERT. The final dataset included 1.4 million reports, of which 15.4% were labeled as `SERIOUS`.

We used pretrained, frozen embeddings to isolate the contribution of narrative information without end-to-end fine-tuning. This choice improves model transparency and makes the approach more scalable and reproducible for public health use cases. Fine-tuning transformer encoders remains a promising direction for future work.

### D. Label Construction and Structured Feature Setup

To construct the binary outcome label `SERIOUS`, we followed CDC and FDA guidelines, marking a report as serious (SERIOUS=1) if any of the following VAERS flags were set to "Yes": `DIED`, `HOSPITAL`, `L_THREAT`, `DISABLE`, or `BIRTH_DEFECT`. These fields were excluded from input features to prevent label leakage. Reports with all flags blank or marked "No" were labeled as not serious (SERIOUS=0). While birth defects may not always reflect acute escalation, they are federally designated as serious outcomes under VAERS reporting guidelines and are treated accordingly in this work.

Structured features included `AGE_YRS`, `SEX`, and days from vaccination to symptom onset. In ablation experiments, manufacturer features were found to contribute minimally: models trained on manufacturer fields alone performed poorly (F1 = 0.33), and their removal slightly improved hybrid model performance. As a result, our primary models exclude manufacturer identifiers from the structured input.

In some configurations, we included the structured symptom fields (SYMPTOM1–5) alongside demographic and timing variables. Table I reports performance both with and without these fields. However, as illustrated in Figures 4 and 5, SYMPTOM1–5 often omit critical clinical details, such as escalation cues, subjective complaints, or symptom trajectories, that remain present in the full narrative. For example, terms like "cardiac arrest" or "unresponsive" were frequently absent from coded fields but recovered through text embeddings. Narrative representations thus provide complementary signal, helping to mitigate sparsity, delay, and imprecision in structured symptom coding.

### E. Risk Stratification by Vaccine Brand

Each report was linked to a primary vaccine brand using the `VAERSVAX` table. For multi-product reports, we followed CDC practice by selecting the first listed vaccine. Manufacturer names were standardized and grouped into categories like Pfizer, Moderna, Janssen, Influenza, and HPV.

Using SBERT-based clusters, we computed the proportion of serious outcomes (`SERIOUS=1`) within each symptom cluster for each vaccine brand. Although clustering was also performed using BioBERT and ClinicalBERT, we selected SBERT for stratification due to its clearer clinical alignment and faster computation.

Cluster assignments were one-hot encoded and used as additional features in the XGBoost classifier. This allowed the model to incorporate latent narrative patterns during classification.

This analysis supports exploratory identification of brand-specific symptom profiles and complements the classification results by highlighting possible variation in how adverse events manifest across vaccine types.

## IV. EVALUATION AND ANALYSIS

### A. Classification Performance

We trained a series of XGBoost classifiers to predict serious adverse events using 5-fold stratified cross-validation. Input configurations included structured-only features, narrative embeddings alone, and hybrid models combining both.

Narrative-only models used sentence-level embeddings from SBERT, BioBERT, or ClinicalBERT. Structured features included patient age, sex, and days to onset. Manufacturer fields were excluded from primary models due to low utility but evaluated separately as a control.

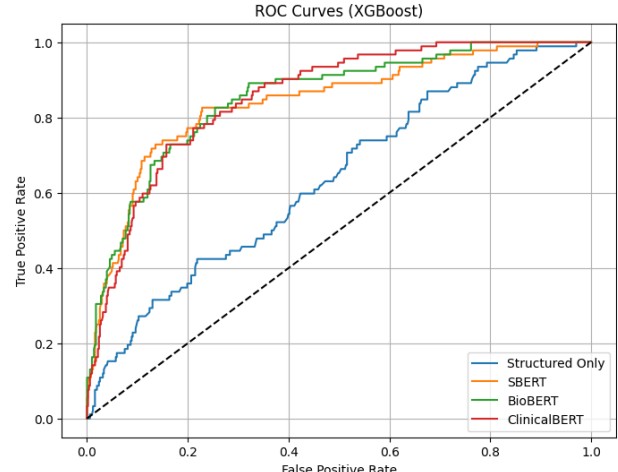

Fig. 2: ROC curves for structured-only and hybrid models. SBERT and BioBERT show similar high performance; the structured-only model underperforms across the curve.

TABLE I: 5-fold cross-validation performance across model configurations. Narrative embeddings improve over structured-only models. Hybrid models perform best.

| Model | AUC | Precision | Recall | F1 |
|---|---|---|---|---|
| Structured (Str.) Only | 0.6891 | 0.2796 | 0.4932 | 0.3569 |
| Manufacturer Only | 0.6012 | 0.2051 | 0.3684 | 0.2659 |
| SBERT Only | 0.9274 | 0.5913 | 0.7011 | 0.6442 |
| BioBERT Only | 0.9259 | 0.5867 | 0.7109 | 0.6418 |
| ClinicalBERT Only | 0.9210 | 0.5805 | 0.6932 | 0.6319 |
| Str. + SBERT | **0.9308** | **0.6138** | 0.7094 | **0.6582** |
| Str. + BioBERT | 0.9275 | 0.5931 | **0.7283** | 0.6538 |
| Str. + ClinicalBERT | 0.9215 | 0.5894 | 0.6949 | 0.6378 |

As shown in Table I, all embedding-based models outperformed the structured-only baseline. SBERT achieved the highest AUC (0.9308) and F1 (0.6582), followed closely by BioBERT and ClinicalBERT. Notably, even narrative-only models performed substantially better than structured-only, confirming that free-text reports carry strong predictive signal. Manufacturer-only features performed poorly (F1 = 0.27) and contributed little when added to hybrid models. As a result, we excluded them from the final configuration.

Figure 2 shows ROC curves for structured-only and hybrid models. SBERT and BioBERT models showed superior performance across the curve, particularly at low false-positive rates – an essential criterion for real-time safety monitoring.

To assess whether the performance gain from structured features is statistically significant, we conducted a non-parametric bootstrap test over the 5-fold cross-validation F1 scores. The SBERT-only model achieved an F1 score of 0.6442, while the Structured + SBERT model reached 0.6582, for an absolute improvement of +0.014. The bootstrap procedure (10,000 resamples) yielded a 95% confidence interval of +0.0093 to +0.0182, with $p < 0.001$. This confirms that the improvement

is statistically significant and not attributable to random variation across folds.

As a control, we also trained a logistic regression model on structured-only features, which achieved an AUC of 0.70, slightly better than XGBoost on the same input (AUC = 0.6891) but still far below hybrid performance. XGBoost was retained for all final models due to its ability to model non-linear interactions between structured variables and narrative embeddings.

*Out-of-Time Generalization* To evaluate how well the model performs on future data, we conducted a temporally stratified experiment: training on reports prior to January 1, 2021 and testing on reports from 2021–2025. The training set included 551,392 reports (11.2% serious), and the test set included 941,016 reports (13.3% serious). The SBERT+structured model achieved an AUC of 0.93 and F1 score of 0.66 on the held-out test set – comparable to performance under 5-fold CV. These results suggest strong generalizability across evolving vaccine types, narratives, and reporting practices.

### B. Demographic Trends

Serious-event prevalence varied substantially across demographic subgroups. Risk increased sharply with age: among individuals aged 85 and older, nearly 47% of reports were labeled as serious. Younger groups (ages 18–44) had much lower rates, typically under 10%. Females accounted for a larger share of serious reports (17%) compared to males (10%). These patterns were also reflected in model behavior. In post-hoc analysis, the hybrid model achieved higher recall in older adults and female patients, likely due to stronger narrative signal in these subgroups. Text descriptions in high-risk groups often contained clearer escalation cues (e.g., hospitalization, ICU admission), which were captured effectively by the embedding models.

Post-hoc subgroup evaluation further confirmed this pattern: the SBERT + structured model achieved a recall of 0.72 for patients aged 65 and older, compared to 0.65 for those under 65. Recall was also higher in female patients (0.70) than in males (0.63), indicating that both clinical severity and demographic variation were reflected in model predictions. Structured features (age, sex, onset delay) improve subgroup recall when combined with embeddings: Age ≥65: 0.72 vs. <65: 0.65; Females: 0.70 vs. Males: 0.63 (Figure 3).

### C. Narrative Interpretability via Token Perturbation

To assess whether the model relies on clinically meaningful terms in the narrative, we conducted a targeted perturbation study. Specifically, we manually masked escalation-related keywords (e.g., "ICU," "unresponsive," "hospitalized") in a sample of correctly classified SERIOUS VAERS reports and measured the change in predicted probability of a serious outcome. As shown in Table II, removing these tokens led to substantial drops in model confidence, indicating that the narrative embeddings capture critical clinical information not present in structured fields.

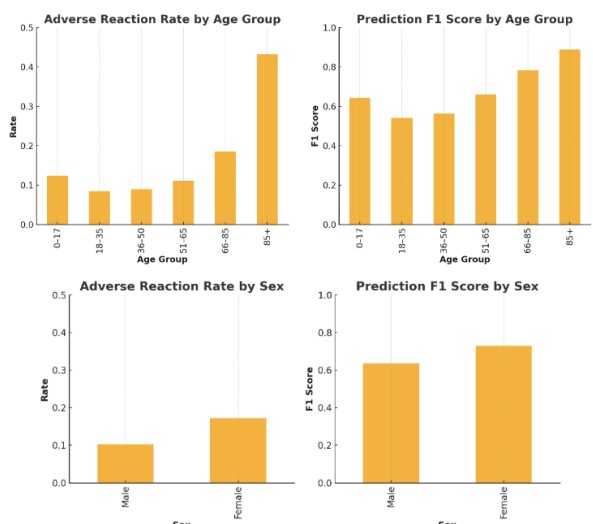

Fig. 3: Demographic patterns in VAERS. (Top) Adverse reaction rates and prediction F1 scores by age group. (Bottom) Rates and F1 scores by sex. The model maintains high performance across subgroups, with elevated risk and predictability among older adults and females.

TABLE II: Effect of masking escalation terms in sample narratives. Removing key words leads to significant drops in serious-event probability.

| Narrative Excerpt | Masked Term | Δ Prob (SERIOUS) |
|---|---|---|
| Admitted to ICU with seizure-like activity | ICU | ↓ 0.38 |
| Found unresponsive at home | unresponsive | ↓ 0.41 |
| Hospitalized for respiratory distress | hospitalized | ↓ 0.36 |

This experiment evaluates predictive influence, not clinical salience. Terms such as "ICU" and "hospitalized" function as escalation proxies rather than diagnostic indicators. Removing clinical descriptors like "seizure" also reduced predicted probability, though to a lesser extent than escalation terms. These results suggest that the model integrates both clinical and contextual cues but disproportionately relies on language associated with event severity. These signals are correlational and do not reflect clinical reasoning.

TABLE III: Comparison of Structured (Str.)-only models with incremental additions: manufacturer fields, symptom fields, and narrative embeddings. Embeddings yield the largest performance gains.

| Model | AUC | Precision | Recall | F1 Score |
|---|---|---|---|---|
| Str. Only | 0.69 | 0.28 | 0.49 | 0.36 |
| Str. + Manufacturer | 0.68 | 0.27 | 0.48 | 0.35 |
| Str. + SYMPTOM1–5 | 0.71 | 0.31 | 0.52 | 0.39 |
| Str. + SBERT | 0.93 | 0.61 | 0.71 | 0.66 |

TABLE IV: Examples where structured fields missed key clinical events captured in narrative text. These cases were misclassified by the structured-only model and correctly flagged by SBERT, demonstrating the added value of narrative embeddings.

| VAERS ID | Structured Symptoms | Narrative Excerpt |
|---|---|---|
| 1140026 | *acute kidney injury, asthenia* | "Profound weakness... slid to floor, unable to get up... rhabdomyolysis with acute kidney injury." |
| 994265 | *colitis, amnesia* | "...started having delusions, almost like having a seizure... magnesium was low." |
| 1995556 | *cardiac arrest, dry eye* | "Takotsubo syndrome... inflammatory markers elevated... uterine dilation and curettage." |

TABLE V: Themes identified in SBERT-based symptom clusters. Labels were derived from TF-IDF terms and representative narratives.

| Cluster | Symptom Theme |
|---|---|
| 0 | Febrile and flu-like symptoms (fever, chills, headache) |
| 1 | COVID-related complications (vaccine or infection) |
| 2 | Administration or dosing errors |
| 3 | Allergic skin reactions (rash, hives, itching) |
| 4 | Local injection site reactions |
| 5 | Shingles and herpes zoster symptoms |
| 6 | Acute systemic reactions (ER visits, respiratory distress) |
| 7 | General post-vaccine symptoms (fatigue, pain, dizziness) |
| 8 | Neurological and cardiovascular symptoms (tinnitus, chest pain) |
| 9 | Nonspecific or administrative reports |

### D. Error Analysis and Model Complementarity

We reviewed 200 false negatives from the structured-only model that were correctly classified by the SBERT-enhanced model. In 78% of these cases, critical information was present only in the narrative text.

Key gaps in structured fields included: (1) Missing diagnoses: Terms like "seizure," "stroke," or "Takotsubo syndrome" appeared only in narratives; (2) Temporal/severity cues: Phrases like "collapsed," "gradually worsened," or "admitted to ER" indicated escalation; (3) Subjective language: Descriptions such as "couldn't get up" or "felt strange in spine" carried clinical relevance but were not coded. Table IV demonstrates that embedding models recover semantic detail lost in coding, which is critical for identifying serious outcomes early.

We further examined errors made by the unimodal models. Figure 4 shows examples where the structured-only model failed, but both SBERT-only and combined models succeeded. Figure 5 shows the reverse: SBERT-only errors that were corrected by structured inputs in the combined model. These examples confirm that structured and narrative modalities offer complementary signals.

### E. SBERT-Based Clustering

For clustering, we used 650,000 randomly sampled narrative reports embedded with SBERT. This sample was selected to balance computational feasibility with sufficient coverage of symptom diversity. Full-scale clustering of all 1.4 million reports would have significantly increased memory and processing costs, with limited expected gains in cluster quality or clinical interpretability. KMeans clustering (k = 10) revealed distinct groupings of symptom types (see Table V). These cluster labels were one-hot encoded and tested as classifier features in follow-up experiments, but yielded only marginal gains. They were not used in the primary models reported in Table I.

As shown in Figure 6, symptom severity was uneven across clusters. Cluster 6 (systemic reactions) showed the highest proportion of serious events, while Cluster 9 (administrative reports) showed the lowest.

Clusters aligned reasonably with clinical categories, though overlapping phrasing limited separation – a common issue in high-dimensional text. Clustering was used purely for exploratory analysis, not for classification or model decisions.

*Predictive Contribution.* To assess the value of clustering, we trained ablated models with and without cluster features. Performance deltas were modest but positive: adding cluster labels improved F1 by +0.01 and AUC by +0.005. While not the most influential features, cluster assignments contributed complementary signal, especially for rare or ambiguous cases.

### F. Brand-Level Symptom Patterns

We stratified symptom clusters by vaccine brand, assigning manufacturer based on the first product listed in multi-vaccine reports. Figure 7 shows brand distributions: COVID-19 mRNA vaccines (Pfizer, Moderna) dominate clusters related to injection site and systemic reactions; Shingrix is concentrated in shingles-related clusters; older vaccines appear in administration-related clusters. These patterns are descriptive and not causal. Cluster differences may reflect reporting style, rollout timing, or population factors. Still, narrative clustering can support exploratory analysis of brand-specific symptom distributions without implying differential safety.

## V. DISCUSSION

This study demonstrates that integrating narrative embeddings with structured metadata significantly improves the detection of serious vaccine adverse events. The SBERT+structured model outperforms a structured-only baseline across all key metrics – recall, precision, F1 score, and AUC – highlighting the added value of unstructured symptom narratives. These improvements are especially relevant in public health surveillance, where identifying high-risk cases early must be balanced against minimizing false positives.

Perturbation experiments confirm that narrative embeddings capture medically salient signals absent from structured fields. Masking escalation terms like "ICU" or "unresponsive" caused sharp drops in predicted SERIOUS probability, indicating the model attends to clinically meaningful cues embedded in free text. These findings suggest that clinical narratives encode critical diagnostic information, severity indicators, and symptom progression—often incompletely captured by structured fields such as `SYMPTOM1--5`.

Error analysis supports this: 78% of false negatives from the structured-only model were correctly reclassified using SBERT. In many cases, the structured codes missed critical diagnoses (e.g., stroke, rhabdomyolysis), severity signals (e.g.,

Structured_Misses_Narrative_Catches

| VAERS ID | Structured Symptoms | Narrative Excerpt | Gold | Pred (Struct) | Pred (SBERT) |
|---|---|---|---|---|---|
| 38196 | Diarrhoea, Gastroenteritis, Rash, Vomiting | 24OCT91 normal exam, 1st DTP/OPV/HIB administered; 28OCT91 seen in office for gastroenteritis & acneform rash; 30OCT91 mom called—diarrhea continues, now vomiting; sent to ER; 31OCT91 pt died | 1 | 0 | 1 |
| 38482 | Arthralgia, Face oedema, Oedema peripheral, Pyrexia, Rash | Pt received vax 9DEC91 & on 10DEC91 had rash on face spreading, achy joints, fever; admitted to ICU due to swelling of eyes, fingers & feet | 1 | 0 | 1 |
| 38733 | Malaise, Right ventricular failure, Somnolence, Vomiting | 1 day post-flu vax: sleepiness & malaise; vomiting; pt died on 17OCT91; COD: congestive heart failure | 1 | 0 | 1 |
| 38734 | Chest pain, Dyspnoea, Haematemesis, Hyperhidrosis, Pallor | Within hours of flu vax: vomiting with blood, chest pain, SOB, pallor; hospitalized & died on 15OCT91; COD: congestive heart failure | 1 | 0 | 1 |
| 38761 | Agitation, Diarrhoea, Stupor | 3 days after immunization: cranky, diarrhea; found unresponsive in crib; resuscitation failed | 1 | 0 | 1 |

Fig. 4: Examples of structured-only model failures that are correctly classified by the SBERT-only and combined models. Narrative embeddings captured escalation signals (e.g., ICU admission, death) that were missing from structured fields.

Narrative_Misses__Structured_Catches

| VAERS ID | Structured Symptoms | Narrative Excerpt | Gold | Pred (Struct) | Pred (SBERT) |
|---|---|---|---|---|---|
| 38222 | Cardiac arrest, Coma, Hypotension | Pt was listless and tired; taken to ER and found in cardiac arrest. | 1 | 1 | 0 |
| 38591 | Seizure, Cyanosis, Respiratory arrest | Seen post-vax for seizure, lips blue, needed resuscitation. | 1 | 1 | 0 |
| 38674 | Anaphylaxis, Shock, Rash generalized | After vax, broke out in rash, seemed tired; pt collapsed at home. | 1 | 1 | 0 |
| 38841 | Bradycardia, Hypoxia, Cardiac failure | Flu-like illness reported, then found unresponsive with low HR. | 1 | 1 | 0 |
| 38960 | Convulsion, Apnea, Tachycardia | Feverish but alert after vax; later developed apnea and seizures. | 1 | 1 | 0 |

Fig. 5: Examples of SBERT-only model failures that are correctly classified by the combined model, based on signal from structured fields. These illustrate the limitations of relying solely on narrative embeddings and highlight modality complementarity.

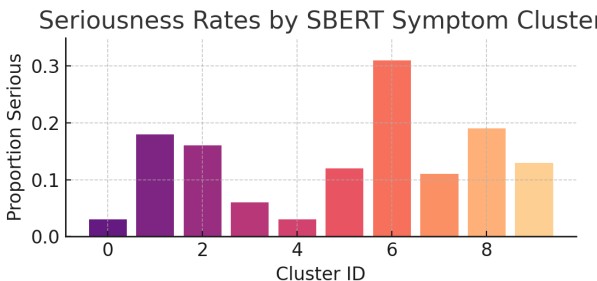

Fig. 6: Proportion of serious outcomes across symptom clusters. Clusters associated with systemic reactions (e.g., Cluster 6) had the highest severity rates.

collapse, ER admission), or subjective symptoms that conveyed clinical urgency. These examples show that pretrained language models help recover meaningful information often lost in traditional coding systems.

Beyond classification, clustering SBERT embeddings revealed symptom groupings that were clinically interpretable and aligned with outcome severity. Some clusters, such as those involving systemic reactions, were enriched for hospitalizations and life-threatening cases, while others reflected mild or administrative reports. While ClinicalBERT produced tighter vector clusters, SBERT yielded more clinically distinct groupings on manual review, justifying its use in subsequent analyses.

Stratifying symptom clusters by manufacturer revealed differences across products (e.g., mRNA vaccines appeared in more systemic clusters). These exploratory patterns may reflect confounding and were not used in classification or decision-making. These results reinforce the benefit of hybrid modeling for both accuracy and transparency in surveillance tasks.

*A. Limitations*

This work has several limitations. VAERS is a passive surveillance system based on voluntary reporting, which can introduce bias, missing data, and underreporting.

Narrative quality is highly variable, and structured fields may be incomplete or inconsistently coded. The outcome label `SERIOUS` is based on administrative reporting, not clinical adjudication. Additionally, text embeddings may capture stylistic patterns or demographic signals unrelated to health status. Approximately 24% of reports were excluded during preprocessing due to brevity or duplication. This may disproportionately exclude older vaccine reports or specific populations and should be considered when interpreting generalizability. Observed associations between symptom clusters and vaccine brands are exploratory and may reflect confounding. These findings are intended to support hypothesis generation, not to imply causal relationships.

Finally, while cross-validation provides a strong internal estimate of model performance, it does not test how well the model generalizes to future data. Because vaccine formulations, public health policies, and reporting language change over time, model performance may vary across reporting periods. Our temporal generalization results (Section IV-A) demonstrate robustness over time. Future work could explore seasonal or variant-specific trends.

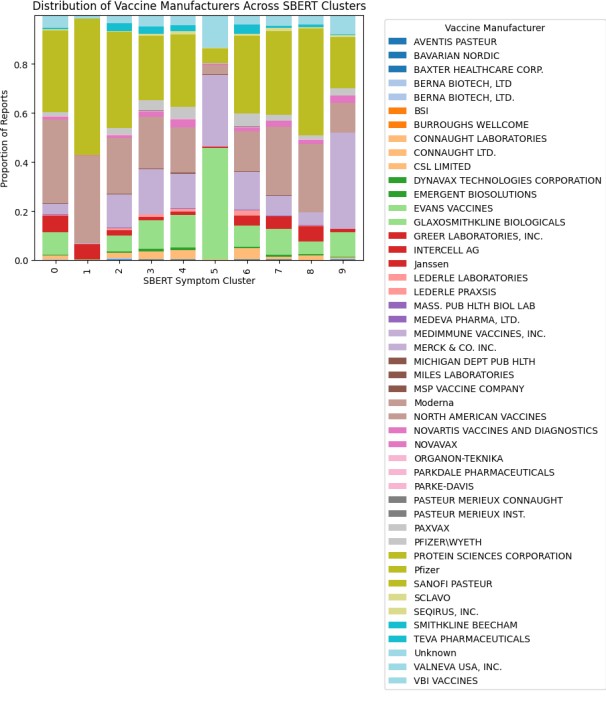

Fig. 7: Distribution of vaccine manufacturers across symptom clusters. COVID-19 mRNA vaccines dominate certain clusters (e.g., local reactions, systemic symptoms). Non-COVID vaccines appear in shingles- and error-related clusters.

*B. Ethics*

Models used for vaccine surveillance must be rigorously validated to avoid reinforcing disparities or missing true safety signals. Misclassifications could delay care or undermine public trust. Fair, interpretable models, combined with human oversight, are essential for responsible deployment in clinical and public health settings. Fairness evaluation remains future work. We also emphasize the need for reproducibility and transparency in post-market surveillance tools. Code and model configurations used in this study are available here: github.com/jonfeld/vaers_modeling/

## VI. CONCLUSION

We developed a hybrid framework combining structured metadata and narrative embeddings to improve detection of serious vaccine adverse events. The model enhances both predictive accuracy and interpretability, recovering context often missing from structured fields alone. Future work should explore booster effects, newer vaccines, and real-time deployments. A comprehensive fairness audit across demographic and linguistic groups remains essential. Additionally, benchmarking against emerging narrative-only baselines like DAE-DRA, once the model or an inference API becomes publicly available, will help clarify the tradeoffs between integrated and text-only approaches.

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
