# OpenReview forum: "Hybrid Modeling of Serious Vaccine Adverse Events Using Narrative Embeddings and Structured Data"
_IEEE.org/EMBS/BHI/2025/Conference — BHI 2025_

### Official Review · Reviewer_TdaS · 2025-07-13
**Hybrid modeling: Structured data + embeddings**

**Confidence:** 4
**Clarity Of Writing:** good
**Clinical Significance:** great
**Methodological Novelty:** good
**Overall Rating:** 4
**Final Rating:** 7

**Experiments And Results:**

fair

**Questions For The Authors:**

Refer to previous section

**Strengths:**

1.	The paper is well written, with a clear structure and logical flow.
2.	The embedding-based clustering analysis is insightful and adds value to the interpretation of the narrative data.
3.	The paper is pretty transparent about the limitations, which is commendable.
4.	The inclusion of an ethics section is appreciated and reflects awareness of the broader implications of the work.

**Summary Of The Paper:**

The manuscript presents a model for classifying VAERS (Vaccine Adverse Event Reporting System) events as serious or non-serious by combining free-text patient narratives with structured data (e.g., age, sex). For the textual component, the authors use pretrained transformer models such as SBERT, BioBERT, and ClinicalBERT to generate embeddings. These embeddings are then combined with structured features and passed to an XGBoost classifier. The paper also employs SHAP to interpret the model's predictions and reports improved performance using this combined approach.

**Weaknesses:**

1. The manuscript would benefit from EDA, including cross plots showing seriousness by age, sex, and other structured variables, as well as sample distribution of serious vs. non-serious cases. This contextual information is essential for interpreting model performance and understanding dataset characteristics.

2. It remains unclear how much predictive power the structured features actually contribute. While the authors state that both embeddings and structured data appear important in SHAP plots, structured variables like Age do not visibly feature in the plots shown.

3. Could the authors report model performance using only the narrative embeddings, excluding structured data, to assess the relative contribution of each data source?

4. An ablation study systematically removing structured features one at a time would provide a clearer understanding of their importance.

5.  While SHAP is a powerful interpretability tool, the current use—applying SHAP to embedding dimensions—raises concerns. It is not clear what an "important" embedding dimension represents semantically. A more meaningful approach might involve:
Applying SHAP at the text input level, perturbing narrative text tokens (e.g., by removing or substituting them), computing their embeddings, and observing the impact on model output.
This can be operationalized by treating the model as two components (1) text embedding, and (2) classification: which SHAP can still treat as a single model due to its model-agnostic nature. This alternative would allow for token-level attribution, which is likely to be more interpretable and less prone to misinterpretation than attributing importance to abstract embedding dimensions.

---

### Official Review · Reviewer_Fdoi · 2025-07-16
**A ML prediction model of serious adverse vaccine events using structured attributes and embedded narrative vectors**

**Confidence:** 4
**Clarity Of Writing:** good
**Clinical Significance:** good
**Methodological Novelty:** fair
**Overall Rating:** 6
**Final Rating:** 6

**Experiments And Results:**

fair

**Questions For The Authors:**

1. It is unclear which structured data was used in the experiment, especially the Coded symptom fields, which play a key connection with outcome of the events. Further experiments should be performed to see the effect Coded SYMPTOM fields. That is, structured data + Coded SYMPTOM1-5 vs structured data + symptom embeddings.
2. Why XGBoost is chosen as the prediction model? As the authors stated, logistic regression exhibited better AUC than XGBoost. Interpretation capability seems to be not a good reason, since SHAP is applicable to any ML models.
3. The SHAP values of structured information such as AGE_YEARS are missing in Fig 3, which is inconsistent with description in the caption.
4. VAERS data is temporal in nature. Using k-fold cross validation rather than temporal prediction seems problematic.

**Strengths:**

The main findings show the importance of incorporating unstructured data, e.g., symptom narrative in this study, with structured data for machine learning prediction, even for unclean medical data like VAERS. The main novelty of the proposed method is using a transform (embedding vector) of narrative from some pre-trained language models. Experimental results also show the proposed method outperform the baseline without narrative embeddings.

**Summary Of The Paper:**

This paper proposes a machine learning-based prediction of serious adverse vaccine events (AVEs), which considers both structured data and unstructured narrative. The proposed method is tailored to the VAERS data. Three different sentence embedding models, SBERT, BioBERT, and ClinicalBERT were used to generate embedded vector of SYMPTOM_TEXT in VAERS. Then, a processed dataset of VAERS from 1990 to 2025 composed of structured data, AGE_YEARS, SEX, NUMDAYS, as well as the embedded narrative vector, was used to train a XGBoost-based prediction of serious AVEs. Experimental results show the proposed hybrid method is superior to baseline model using only structured data.

**Weaknesses:**

1. The advantage of combining structured data and unstructured data for machine learning is well known in different application domains, due to rich information available for training the model. Although this work focused on serious vaccine event prediction, the proposed approach exhibits not too much technical innovation.
2. Another concern is why excluding the recent model DAEDRA dedicate to vaccine symptom narrative in the compared models.
3. VAERS contains official coded symptoms (a more precise and correct information) extracted from the original narrative. How a ML model trained from the demographic structured data along with the coded symptoms perform is unclear in this paper.

---

### Official Review · Reviewer_uhv4 · 2025-07-16
**Vaccine adverse event prediction using pretrained text encoder and additional features**

**Confidence:** 3
**Clarity Of Writing:** great
**Clinical Significance:** good
**Methodological Novelty:** good
**Overall Rating:** 7

**Experiments And Results:**

good

**Questions For The Authors:**

-In Fig. 1, is the cluster formed as a result of k-mean has any syntax meaning?
- Is there any overlap between VAERS and the training data for selected language models?

**Strengths:**

good dataset curation - could be beneficial to research community and posses clinical impacts

**Summary Of The Paper:**

This paper presents a prediction pipeline for vaccine adverse event using embeddings obtained from pre-trained language model. The text encoder takes structured metadata, paired with crafted feature vectors to jointly perform prediction using XGBoost. Experimental results show that SBERT performs best overall, suggesting narrative text enhance prediction outcomes.

**Weaknesses:**

- the contribution of cluster assignments to overall detection performance is under explored without a comparison between AUC/Precision/Recall/F1 with or without such extra info

---

### Official Review · Reviewer_NpY8 · 2025-07-16
**Compelling idea to explore serious vaccine adverse events**

**Confidence:** 5
**Clarity Of Writing:** excellent
**Clinical Significance:** excellent
**Methodological Novelty:** great
**Overall Rating:** 7

**Experiments And Results:**

excellent

**Questions For The Authors:**

Can you merge similar symptoms using your clustering methods or other dimensionality reduction methods? Additionally, can you consider adding a fairness analysis aspect to assess bias or variability in the narrative text to determine whether it affects performance?

**Strengths:**

The paper effectively motivates the need to include data beyond structured variables by highlighting the gaps in symptom coding. The authors conduct an insightful error analysis by showing that many false negatives from using only structured data contain critical information in the narrative text. This is essential for their argument that using embeddings are useful with compelling qualitative support.

**Summary Of The Paper:**

This paper “Hybrid Modeling of Serious Vaccine Adverse Events Using Narrative Embeddings and Structured Data” introduces a machine learning framework integrating structured metadata and unstructured narrative text from VAERS reports to predict serious vaccine adverse events. Using sentence embeddings from BioBERT, SBERT, and ClinicalBERT with XGBoost classifiers, the study showcases high accuracy over models that only learn from structured data. The paper also performs clustering on narrative text to uncover clinically relevant patterns and ultimately shows the potential of unstructured narrative text for vaccine safety.

**Weaknesses:**

The binary outcome label used to predict a “serious” event could use more justification. For example, the inclusion of birth defects can be confusing as a marker for a serious adverse event since it is unclear whether it is a direct consequence of the vaccination itself and might end up being a confounder. Additionally, while the narrative text is central to the model’s success, the paper does not address how stylistic differences could affect embeddings and performance. I was also curious whether the authors tried combining clusters to reduce embedding redundancy, which could help reduce noise or improve the generalizability of the model.